# Efficient Differentiable Discovery of Causal Order

## Abstract

In the algorithm Intersort, Chevalley et al. (2024) proposed a score-based method to discover the causal order of variables in a Directed Acyclic Graph (DAG) model, leveraging interventional data to outperform existing methods. However, as a score-based method over the permutahedron, Intersort is computationally expensive and non-differentiable, limiting its ability to be utilised in problems involving large-scale datasets, such as those in genomics and climate models, or to be integrated into end-to-end gradient-based learning frameworks. We address this limitation by reformulating Intersort using differentiable sorting and ranking techniques. Our approach enables scalable and differentiable optimization of causal orderings, allowing the continuous score function to be incorporated as a regularizer in downstream tasks. Empirical results demonstrate that causal discovery algorithms benefit significantly from regularizing on the causal order, underscoring the effectiveness of our method. Our work opens the door to efficiently incorporating regularization for causal order into the training of differentiable models and thereby addresses a long-standing limitation of purely associational supervised learning.

## 1    Introduction

Causal discovery is fundamental for understanding complex systems by identifying underlying causal relationships from data. It has significant applications across various fields, including biology (Meinshausen et al., 2016; Chevalley et al., 2022; 2023), medicine (Feuerriegel et al., 2024), and social sciences (Imbens & Rubin, 2015), where causal insights inform decision-making and advance scientific knowledge. Traditionally, causal discovery has relied heavily on observational data due to the practical challenges and costs associated with conducting large-scale interventional experiments. However, observational data alone often necessitates strong assumptions about the data distribution to ensure identifiability beyond the Markov equivalence class (Spirtes et al., 2000; Shimizu et al., 2006; Hoyer et al., 2008).

The emergence of large-scale interventional datasets, particularly in domains like single-cell genomics (Replogle et al., 2022; Datlinger et al., 2017; Dixit et al., 2016), introduces new opportunities and challenges for causal discovery. Interventional data, obtained through targeted manipulations of variables, offers a unique perspective by revealing causal mechanisms that may be obscured in purely observational studies. Recently (Chevalley et al., 2024) introduced Intersort, which uses the notion of *interventional faithfulness* and a score on causal orders, enabling the inference of causal orderings by comparing marginal distributions across observational and interventional settings.

Despite its promising performance, Intersort faces significant limitations. Specifically, it lacks differentiability, which hinders its integration into gradient-based learning frameworks commonly used in modern machine learning. Additionally, scalability remains a challenge, making this method impractical for applications involving large numbers of variables—a common scenario in fields such as genomics (Replogle et al., 2022; Chevalley et al., 2022) and neuroscience that involve up to tens of thousands of variables.

In this work, we address these challenges by extending Intersort to a continuous realm to make it both differentiable and scalable. We achieve this by expressing the score in terms of a potential function and utilizing differentiable sorting and ranking techniques, including the Sinkhorn operator (Cuturi, 2013). This novel formulation allows us to incorporate a useful inductive bias into downstream tasks as a differentiable constraint or regularizer, enabling seamless integration into gradient-based optimization frameworks.

We develop a causal discovery algorithm that integrates differentiable Intersort score into its objective function. Our empirical evaluations on diverse simulated datasets—including linear, random Fourier features, gene regulatory networks (GRNs) and neural network models—demonstrate that the proposed regularized algorithm significantly outperforms baseline methods such as GIES (Hauser & Bühlmann, 2012) and DCDI (Brouillard et al., 2020) on RFF and GRN data. Moreover, we demonstrate that our approach exhibits robustness across different data distributions and noise types. The algorithm efficiently scales with large datasets, maintaining consistent performance regardless of data size.

Our contributions pave the way for fully leveraging interventional data in various causal tasks in a scalable and differentiable manner. By addressing the limitations of existing methods, we enable the application of interventional faithfulness in large-scale settings and facilitate its integration into modern causal machine learning pipelines.

## 2 RELATED WORK

Causal discovery is a fundamental problem in machine learning and statistics, aiming to uncover causal relationships from data. Traditional methods can be broadly categorized into constraint-based, score-based, and functional approaches. Algorithms like PC (Spirtes et al., 2000) and FCI (Spirtes, 2001) use conditional independence tests to infer causal structures. These methods often require faithfulness and causal sufficiency assumptions, which may not hold in real-world scenarios. Score-based approaches, such as GES (Chickering, 2002) and GIES (Hauser & Bühlmann, 2012), search over possible graph structures to maximize a goodness-of-fit score. More recently, Zheng et al. (2018) introduced NOTEARS, a differentiable approach that formulates causal discovery as a continuous optimization problem using an acyclicity constraint. Most recently proposed models follow the idea of NOTEARS and are thus continuously differentiable, e.g. using neural networks to model the functional relationships (Brouillard et al., 2020; Lachapelle et al., 2019).

Chevalley et al. (2024) recently introduced Intersort, a score-based algorithm to derive the causal order of the variables when many single-variable interventions are available. Intersort relies on a light assumption on the changes in marginal distributions across observational and interventional settings, as measure by a statistical distance (see definition 3 subsequently). Intersort is a two-step algorithm, where the first step, SORTRANKING, finds an initial ordering by taking into account the scale of the measured distances, and the second step, LOCALSEARCH, search in a close neighbourhood in permutation space to improve the score. Their algorithm suffers from scalability issues, as the second step, LOCALSEARCH, becomes computationally intractable for large datasets. Moreover, it is unclear how this inductive bias can be applied to downstream causal task, as the score is not differentiable with respect to a causal order and thus it cannot be integrated into continuously differentiable models. Previous to Intersort, most methods to infer the causal order of a system worked primarily on observational data. For example, EASE (Gnecco et al., 2021) leverages extreme values to identify the causal order. Reisach et al. (2021) observed that the performance of continuously differentiable causal discovery models on synthetic datasets strongly correlates with *varsortability*, which measures to what extent the marginal variance corresponds to the true causal order. Similar ideas were consequently developed to derive the causal order based on score matching (Rolland et al., 2022; Montagna et al., 2023a;b).

# 3 METHOD

## 3.1 DEFINITIONS AND ASSUMPTIONS

In this section, we introduce notations and definitions that are used throughout the paper inspired by (Pearl, 2009; Peters et al., 2017).

Let $(\mathcal{M}, d)$ be a metric space, and let $\mathcal{P}(\mathcal{M})$ denote the set of probability measures over $\mathcal{M}$. We define $D$ to be a statistical distance function $D : \mathcal{P}(\mathcal{M}) \times \mathcal{P}(\mathcal{M}) \to [0, \infty)$ that measures the divergence between probability distributions on $\mathcal{M}$. Consider a set of $d$ random variables $\mathbf{X} = (X_1, X_2, \ldots, X_d)$ indexed by $V = \{1, 2, \ldots, d\}$, with joint distribution $P_{\mathbf{X}}$. We denote the marginal distribution of each variable as $P_{X_i}$ for $i \in V$. A causal graph is a tuple $\mathcal{G} = (V, E)$ of nodes and edges that form a Directed Acyclic Graph (DAG), where $V$ is the set of nodes (variables), and $E \subseteq V \times V$ is the set of directed edges representing causal relationships. An edge $(i, j) \in E$ indicates that variable $X_i$ is a direct cause of variable $X_j$. Let $\mathbf{A}^{\mathcal{G}}$ be the adjacency matrix of $\mathcal{G}$, where $\mathbf{A}^{\mathcal{G}}_{ij} = 1$ if $(i, j) \in E$, and $\mathbf{A}^{\mathcal{G}}_{ij} = 0$ otherwise. For each node $j \in V$, the set of parents $\mathrm{Pa}(j)$ consists of all nodes with edges pointing to $j$, i.e., $\mathrm{Pa}(j) = \{i \in V \mid (i, j) \in E\}$. We denote the set of descendants of node $i$ as $\mathrm{De}_{\mathcal{G}}(i)$, which includes all nodes reachable from $i$ via directed paths. Similarly, the set of ancestors of $i$ is denoted as $\mathrm{An}_{\mathcal{G}}(i)$.

An SCM $\mathcal{C} = (\mathbf{S}, P_N)$ consists of a set of structural assignments $\mathbf{S}$ and a joint distribution over exogenous noise variables $P_N$. Each variable $X_j$ is assigned via a structural equation:

$$X_j = f_j \left( \mathbf{X}_{\mathrm{Pa}(j)}, N_j \right),$$

where $N_j$ is an exogenous noise variable, and $\mathbf{X}_{\mathrm{Pa}(j)}$ are the parent variables of $X_j$.

In our work, we focus on interventions that modify the structural assignments of certain variables. Specifically, we consider interventions where the structural assignment of a variable $X_k$ is replaced by a new exogenous variable $\tilde{N}_k$, independent of its parents $X_k = \tilde{N}_k$.

**Definition 1.** A *causal order* is a permutation $\pi : V \to \{1, 2, \ldots, d\}$ such that for any edge $(i, j) \in E$, we have $\pi(i) < \pi(j)$. This ensures that causes precede their effects in the ordering (Peters et al., 2017).

Since $\mathcal{G}$ is acyclic, at least one causal order exists, though it may not be unique. We denote the set of all causal orders consistent with $\mathcal{G}$ as $\Pi^*$.

**Definition 2.** To measure the discrepancy between a proposed permutation $\pi$ and the true causal graph $\mathcal{G}$, we use the *top order divergence* (Rolland et al., 2022), defined as:

$$D_{\mathrm{top}}(\mathcal{G}, \pi) = \sum_{\pi(i) > \pi(j)} \mathbf{A}^{\mathcal{G}}_{ij}.$$

This divergence counts the number of edges that are inconsistent with the ordering $\pi$, i.e., edges where the cause appears after the effect in the proposed ordering. For any causal order $\pi^* \in \Pi^*$, we have $D_{\mathrm{top}}(\mathcal{G}, \pi^*) = 0$.

**Assumption 1** (Interventional Faithfulness). *Interventional faithfulness (Chevalley et al., 2024) assumes that all directed paths in the causal graph manifest as significant changes in the distribution under interventions as measured by a statistical distance. Specifically, if intervening on variable $X_i$ leads to a detectable change in the distribution of variable $X_j$, then there must be a directed path from $X_i$ to $X_j$ in the causal graph $\mathcal{G}$. Conversely, if there is no directed path from $X_i$ to $X_j$, then intervening on $X_i$ does not affect the distribution of $X_j$ beyond a significance threshold $\epsilon$.*

Interventional faithfulness allows us to use statistical divergences between observational and interventional distributions to infer the causal ordering of variables. By assuming interventional faithfulness, we can relate changes observed under interventions to the underlying causal structure. More formally, it is defined as:

**Definition 3** (Chevalley et al. (2024)). Given the distributions $P_X^{\mathcal{C},(\emptyset)}$ and $P_X^{\mathcal{C},do(X_k:=\tilde{N}_k)}, \forall k \in \mathcal{I}$, we say that the tuple $(\tilde{N}, \mathcal{C})$ is $\epsilon$-*interventionally faithful* to the graph $\mathcal{G}$ associated to $\mathcal{C}$ if for all $i \neq j, i \in \mathcal{I}, j \in V$, $D\left(P_{X_j}^{\mathcal{C},(\emptyset)}, P_{X_j}^{\mathcal{C},do(X_i:=\tilde{N}_i)}\right) > \epsilon$ if and only if there is a directed path from $i$ to $j$ in $\mathcal{G}$.

### 3.2 Differentiable score

While Intersort demonstrates cutting-edge results in discerning causal order among variables, its primary drawback is the substantial computational cost, which restricts its application to small-scale problems. The authors of the original paper acknowledged this limitation, confining their evaluation to a mere 30 nodes Chevalley et al. (2024). A covariate set of this size is prohibitively small for many real-world problems, such as those in genomics and climate change, where tens of thousands of variables are considered. We aim to enhance the scalability of Intersort through a differentiable objective function. This not only facilitates scaling to a considerably larger number of variables but also enables the integration of this algorithm in end-to-end gradient-based model training. In the subsequent sections, we initially revisit the fundamental score that underpins Intersort. Following this, we proceed to present a differentiable formulation, DiffIntersort, that addresses these shortcomings.

*Intersort score–* Given an observational distribution $P_X^{\mathcal{C},(\emptyset)}$ and a set of interventional distributions $\mathcal{P}_{int} = \{P_X^{\mathcal{C},do(X_k:=\tilde{N}_k)}, k \in \mathcal{I}\}, \mathcal{I} \subseteq V$, Chevalley et al. (2024) define the following score for a permutation $\pi$, for some statistical distance $D : \mathcal{P}(M) \times \mathcal{P}(M) \to [0, \infty), \epsilon > 0, c \geq 0$:

$$S(\pi, \epsilon, D, \mathcal{I}, P_X^{\mathcal{C},(\emptyset)}, \mathcal{P}_{int}, c) = \sum_{\pi(i)<\pi(j), i\in\mathcal{I}, j\in V} \left( D\left(P_{X_j}^{\mathcal{C},(\emptyset)}, P_{X_j}^{\mathcal{C},do(X_i:=\tilde{N}_i)}\right) - \epsilon \right) \\ + c \cdot d \cdot \mathbf{1}_{D\left(P_{X_j}^{\mathcal{C},(\emptyset)}, P_{X_j}^{\mathcal{C},do(X_i:=\tilde{N}_i)}\right)>\epsilon} \tag{1}$$

We parameterize an ordering of the variable as determined by a permutation of the variables $\pi$ via a potential $\boldsymbol{p} \in \mathbb{R}^d$ such that $\pi(i) < \pi(j) \iff p_i > p_j$. We write the permutation matrix associated to $\boldsymbol{p}$ as $\boldsymbol{\sigma}(\boldsymbol{p})$, which is a $d \times d$ binary matrix. Let $s : \mathbb{R}^d \to \mathbb{R}^d$ be the sorting function, which sorts a vector of $d$ numbers in descending order. The Jacobian of $s(\boldsymbol{p})$ with respect to $\boldsymbol{p}$ is equal to the permutation matrix $\boldsymbol{\sigma}(\boldsymbol{p})$. We define $(\text{grad}(\boldsymbol{p}))_{ij} = p_i - p_j$, which is non-negative if and only if $\pi(i) < \pi(j)$ in the associated topological order. Applying the element-wise Step function produces $(\text{Step}(\text{grad}(\boldsymbol{p})))_{ij} = 1_{p_i-p_j>0}$ which is a matrix of the possible edges according to the potential $\boldsymbol{p}$.

We aim to rewrite the score such that it is parameterized by a potential $\boldsymbol{p}$. By building the matrix $\mathbf{D} \in \mathbb{R}^{d \times d}$ as

$$\mathbf{D}_{ij} = \begin{cases} D\left(\left(P_{X_j}^{\mathcal{C},(\emptyset)}, P_{X_j}^{\mathcal{C},do(X_i:=\tilde{N}_i)}\right) - \epsilon\right) + c \cdot d \cdot \mathbf{1}_{D\left(P_{X_j}^{\mathcal{C},(\emptyset)}, P_{X_j}^{\mathcal{C},do(X_i:=\tilde{N}_i)}\right)>\epsilon} & \text{if} \quad i \in \mathcal{I} \\ 0 & \text{if} \quad i \notin \mathcal{I} \end{cases} \tag{2}$$

we can write the score in terms of the potential instead of permutation as follows:

$$S(\boldsymbol{p}, \epsilon, D, \mathcal{I}, P_X^{\mathcal{C},(\emptyset)}, \mathcal{P}_{int}, c) = \sum_{i,j} \left(\mathbf{D} \odot \text{Step}(\text{grad}(\boldsymbol{p}))\right)_{ij}. \tag{3}$$

The relationship between the potential and permutation is clarified through the following theoretical result.

**Theorem 1.** *Let* $\mathbb{P} = \arg\max_{\boldsymbol{p}} S(\boldsymbol{p}, \epsilon, D, \mathcal{I}, P_X^{\mathcal{C},(\emptyset)}, \mathcal{P}_{int}, c)$ *s.t.* $\boldsymbol{p}_i \neq \boldsymbol{p}_j \forall i, j \in \{1, \cdots, d\}$ *be the set of potentials that maximize the score, such that no two potentials are equal.* $\Pi = \arg\max_{\pi} S(\pi, \epsilon, D, \mathcal{I}, P_X^{\mathcal{C},(\emptyset)}, \mathcal{P}_{int}, c)$ *be the set of permutations that maximize the Intersort score. For all* $\pi \in \Pi$, *there is a* $\boldsymbol{p} \in \mathbb{P}$ *such that* $\pi(i) < \pi(j) \iff p_i > p_j$.

The proof can be found in the appendix in section 6.1. This score is still not practically useful as it provides non-informative gradients for $\boldsymbol{p}$. To remedy this, inspired by Annadani et al. (2023) we define $\boldsymbol{L} \in \{0, 1\}^{d \times d}$ as a matrix with upper triangular part to be 1, and vector $\boldsymbol{o} = [1, \ldots, d]$. They propose the formulation

$$\text{Step}(\text{grad}(\boldsymbol{p})) = \boldsymbol{\sigma}(\boldsymbol{p}) \boldsymbol{L} \boldsymbol{\sigma}(\boldsymbol{p})^T \qquad \text{where } \boldsymbol{\sigma}(\boldsymbol{p}) = \arg\max_{\boldsymbol{\sigma}' \in \Sigma_d} \boldsymbol{p}^T(\boldsymbol{\sigma}' \boldsymbol{o}) \qquad (4)$$

where $\boldsymbol{\Sigma}_d$ represents the space of all $d$ dimensional permutation matrices. This formulation opens the door to use tools from the differentiable permutation optimization literature. More specifically, we need to build a smooth approximation to the argmax operator in the definition of $\boldsymbol{\sigma}(\boldsymbol{p})$. Fortunately, theoretical results for this transformation is already known in the optimization literature which is built upon the concept of Sinkhorn Operator that we briefly discuss in the following. We refer readers to the original paper (Sinkhorn, 1964) and further applications (Adams & Zemel, 2011) of Sinkhorn operator for detailed presentation.

The Sinkhorn operator, $\mathcal{S}(\boldsymbol{M})$, on a matrix $\boldsymbol{M}$, involves a sequence of alternating row and column normalizations, known as Sinkhorn iterations. Mena et al. (2018) demonstrated that the non-differentiable $\arg\max$ problem

$$\boldsymbol{\sigma} = \arg\max_{\boldsymbol{\sigma}' \in \Sigma_d} \langle \boldsymbol{\sigma}', \boldsymbol{M} \rangle_F \qquad (5)$$

can be relaxed using an entropy regularizer, where the solution is given by $\mathcal{S}(\boldsymbol{M}/t)$. Specifically, they showed that $\mathcal{S}(\boldsymbol{M}/t) = \arg\max_{\boldsymbol{\sigma}' \in \mathcal{B}_d} \langle \boldsymbol{\sigma}', \boldsymbol{M} \rangle + th(\boldsymbol{\sigma}')$, where $h(\cdot)$ denotes the entropy function. This regularized solution converges to the solution of eq. (5) as $t \to 0$, shown by $\lim_{t \to 0} \mathcal{S}(\boldsymbol{M}/t)$. The implication for our setting is that we can write eq. (4) as

$$\arg\max_{\boldsymbol{\sigma}' \in \Sigma_d} \boldsymbol{p}^T(\boldsymbol{\sigma}' \boldsymbol{o}) = \arg\max_{\boldsymbol{\sigma}' \in \Sigma_d} \langle \boldsymbol{\sigma}', \boldsymbol{p}\boldsymbol{o}^T \rangle_F = \lim_{t \to 0} \mathcal{S}\left(\frac{\boldsymbol{p}\boldsymbol{o}^T}{t}\right). \qquad (6)$$

In practice, we approximate the limit with a value of $t > 0$ and a certain number of iterations $T$, which results in a differentiable and doubly stochastic matrix in the $d$-dimensional Birkhoff polytope $\mathcal{B}_d$. The parameter $t > 0$ acts as a temperature controlling the smoothness of the approximation. In our experiments, we use $t = 0.05$ and $T = 500$. We then apply the Hungarian algorithm Kuhn (1955) to obtain a binary matrix, and use the straight-through estimator in the backward pass. The resulting binary matrix is denoted as $\mathcal{S}_{\text{bin}}^T(\boldsymbol{p}\boldsymbol{o}^T/t)$ with "bin" emphasizing a binary-valued matrix. As a result, the score becomes differentiable and can be differentiated through the iterations of the Sinkhorn operator. By replacing the non-differentiable part of eq. (2) with this matrix, the complete form of the differentiable score (we call it DiffIntersort) is derived as

$$S(\boldsymbol{p}, \epsilon, D, \mathcal{I}, P_X^{\mathcal{C},(\emptyset)}, \mathcal{P}_{int}, t, T) = \sum_{i,j} \left( \mathbf{D} \odot \left( \mathcal{S}_{\text{bin}}^T\left(\frac{\boldsymbol{p}\boldsymbol{o}^T}{t}\right) \boldsymbol{L} \mathcal{S}_{\text{bin}}^T\left(\frac{\boldsymbol{p}\boldsymbol{o}^T}{t}\right)^T \right) \right)_{ij}. \qquad (7)$$

For the rest of the paper, we drop in subscript "bin" and use $S(\mathbf{p})$ for conciseness. The maximizers of the DiffIntersort score and the Intersort score are equal for $t \to 0$ and $T \to \infty$ (Theorem 1). The DiffIntersort score $S(\mathbf{p})$ can be maximized with respect to the potential vector $\mathbf{p}$ using gradient descent algorithms. This allows us to find the ordering of variables that best aligns with the interventional data, according to the statistical distances captured in $\mathbf{D}$.

### 3.3 Causal Discovery Algorithm

After deriving the differentiable score in the previous section, we now proceed to use the score in a causal discovery algorithm. Let's consider a dataset $\mathbf{X} \in \mathbb{R}^{n \times d}$ consisting of $n$ observations of $d$ variables $\{X_1, X_2, \ldots, X_n\}$. We assume the data to be generated from an unknown Structural Equation Model (SEM), which can be described by a Directed Acyclic Graph (DAG) $\mathcal{G}$ representing the causal relationships among variables. Our goal is to recover the causal structure and ordering of the variables from both observational and interventional data. We introduce a potential vector $\mathbf{p} \in \mathbb{R}^d$ that induces a permutation representing the causal ordering of variables. Let $S(\mathbf{p})$ be the DiffIntersort score, which measures the consistency of the ordering induced by $\mathbf{p}$ with the interventional data. The causal discovery can then be formulated as a constrained optimization problem

$$\min_{\theta, \mathbf{p}} \mathcal{L}_{\text{fit}}(\theta, \mathbf{p}) \quad \text{subject to} \quad \mathbf{p} \in \arg\min_{\mathbf{p}'} S(\mathbf{p}') \tag{8}$$

where $\theta$ represents the parameters of the causal mechanisms (e.g., weight matrices in linear models), and $\mathcal{L}_{\text{fit}}(\theta, \mathbf{p})$ is the fitting loss that measures how well the model with parameters $\theta$ explains the observed data. The constraint ensures that the potential vector $\mathbf{p}$ minimizes the DiffIntersort score, thus enforcing a causal ordering consistent with the interventional data. We transform the constrained optimization into an unconstrained penalized problem

$$\min_{\theta, \mathbf{p}} \quad \mathcal{L}_{\text{fit}}(\theta, \mathbf{p}) + \lambda S(\mathbf{p}), \tag{9}$$

where $\lambda > 0$ is a regularization parameter controlling the trade-off between fitting the data and enforcing the causal ordering through the DiffIntersort score.

As an example, a linear causal model can be constructed as

$$X_j = \sum_{i=1}^{d} W_{ji} X_i + b_j + N_j, \tag{10}$$

where $W_{ji}$ are the entries of the weight matrix $\mathbf{W} \in \mathbb{R}^{d \times d}$, $b_j$ is the bias term, and $N_j$ is a noise term. To enforce the causal ordering induced by $\mathbf{p}$, we use the permuted upper-triangular matrix $\boldsymbol{M}_{\mathbf{p}} = \mathcal{S}_{\text{bin}}^T(\boldsymbol{p}\boldsymbol{o}^T/t) \boldsymbol{L} \mathcal{S}_{\text{bin}}^T(\boldsymbol{p}\boldsymbol{o}^T/t)^T$, which is a $d \times d$ matrix with $d(d-1)/2$ entries equal to 1. The matrix represents the possible locations of edges in the graph according to the causal ordering $\mathbf{p}$. By element-wise multiplication $\tilde{\mathbf{W}} = \mathbf{W} \circ \boldsymbol{M}_{\mathbf{p}}^T$, matrix $\boldsymbol{M}_{\mathbf{p}}$ acts as a mask to ensure that variable $X_j$ can only depend on variables preceding it in the causal ordering. The predicted values can be written in terms of the entries of $\tilde{\mathbf{W}}$ as $\hat{X}_j = \sum_{i=1}^{d} \tilde{\mathbf{W}}_{ji} X_i + b_j$.

Inspired by the fitting loss in Shen et al. (2023), we define the fitting loss $\mathcal{L}_{\text{fit}}(\theta)$ as:

$$\mathcal{L}_{\text{fit}}(\theta, \mathbf{p}) = \frac{1}{n^0} \sum_{i=1}^{n^0} \ell(\mathbf{x}_i, \hat{\mathbf{x}}_i; \theta, \mathbf{p}) + \gamma \sum_{e \in \mathcal{E}} \omega^e \left( \frac{1}{n^e} \sum_{i=1}^{n^e} \ell^e(\mathbf{x}_i, \hat{\mathbf{x}}_i; \theta, \mathbf{p}) - \frac{1}{n^0} \sum_{i=1}^{n^0} \ell^0(\mathbf{x}_i, \hat{\mathbf{x}}_i; \theta, \mathbf{p}) \right), \tag{11}$$

where $\ell(\mathbf{x}_i, \hat{\mathbf{x}}_i; \theta)$ is the mean absolute error (MAE) loss function for observational sample $i$, $\ell^e(\mathbf{x}_i, \hat{\mathbf{x}}_i; \theta)$ is the loss for samples in environment $e$. In our case, an environment corresponds to an intervention on one variable. $\gamma \geq 0$ is a parameter controlling the emphasis on invariance across environments. We use $\gamma = 0.5$.

$\omega^e$ are weights for each environment. We set $\omega^e = \frac{1}{|\mathcal{E}|}$. $\mathcal{E}$ is the set of environments, with $0 \in \mathcal{E}$ denoting the reference observational environment. $n^e$ is the number of samples in environment $e \in \mathcal{E}$. The loss encourages the model to fit the data in the reference environment while penalizing deviations in performance across different environments, promoting robustness to interventions. This should encourage the weights to corresponds to the true causal weights, as the equivalence between robustness and causality is well established (Meinshausen, 2018).

Combining the fitting loss and the regularization terms, the final loss function is:

$$
\begin{aligned}
\mathcal{L}(\theta, \mathbf{p}) =& \frac{1}{n^0} \sum_{i=1}^{n^0} \ell(\mathbf{x}_i, \hat{\mathbf{x}}_i; \theta, \mathbf{p}) \\
&+ \gamma \sum_{e \in \mathcal{E}} \omega^e \left( \frac{1}{n^e} \sum_{i=1}^{n^e} \ell^e(\mathbf{x}_i, \hat{\mathbf{x}}_i; \theta, \mathbf{p}) - \frac{1}{n^0} \sum_{i=1}^{n^0} \ell^0(\mathbf{x}_i, \hat{\mathbf{x}}_i; \theta, \mathbf{p}) \right) \\
&+ \lambda_1 \|\mathbf{W}\|_1 + \lambda_2 S(\mathbf{p}).
\end{aligned}
\tag{12}
$$

This loss function includes all the components: (1) *Data Fitting Loss* Measures how well the model predicts the observed data, adjusted for interventions, (2) *Environment Invariance Penalty* Encourages the model to have consistent performance across different environments, (3) $L_1$ *Regularization*: Promotes sparsity in the weight matrix $\mathbf{W}$, (4) *DiffIntersort Regularization*: Incorporates interventional faithfulness by penalizing with the DiffIntersort score $S(\mathbf{p})$ eq. (7) . We also note that no acyclicity constraint is needed as the weight matrix is enforced to be acyclic through the masking based on the causal order $\mathbf{p}$.

## 4 EMPIRICAL RESULTS

We next evaluate the proposed DiffIntersort differentiable score both in its effectiveness in deriving the causal order of a system, as well as it usefulness as a differentiable constraint in a causal discovery model.

We first evaluate the DiffIntersort score in it ability to recover the causal order in simulated graphs and distance matrices. We here reproduce the experiment of (Chevalley et al., 2024). We compare the top order divergence of DiffIntersort to SORTRANKING, and to Intersort for 5 and 30 variables, and the upper-bounds of Thm 2 and Thm 4 derived in (Chevalley et al., 2024). Intersort does not scale beyond 100 variables. The upper-bounds serve as a sanity check to assess how far to the true optimum of the score the approximate solution is. We evaluate on both Erdős-Rényi distribution (Erdős et al., 1960) and scale-free network modelled by the Barabasi-Albert distribution Albert & Barabási (2002), with varying edge densities and intervention coverage. The results are reported in fig. 1 for 2000 variables and in figs. 5 and 6 for 5, 30, 100 and 1000 variables. It is crucial that our score is optimizable up to at least 2000 variables as it is a common scale in real world datasets such as single-cell transcriptomics (Replogle et al., 2022). As observed, DiffIntersort fulfills the upper-bounds for all settings, even at large scale. At large scale, it also outperforms SORTRANKING in almost all settings. Those results validate our proposed approach of solving the Intersort problem in a continuous and differentiable framework, and guarantees that it is not limited by scale.

We now evaluate our method, DiffIntersort, on simulated data and compare its performance to various baseline methods. We follow the experimental setup of Chevalley et al. (2024) to ensure a fair and consistent evaluation across different domains. Specifically, we generate graphs from an Erdős-Rényi distribution (Erdős et al., 1960) with an expected number of edges per variable $c \in \{1, 2\}$. Data is simulated using both linear relationships and random Fourier features (RFF) additive functions to capture non-linear dependencies. In addition to these synthetic datasets, we apply our models to simulated single-cell RNA sequencing data

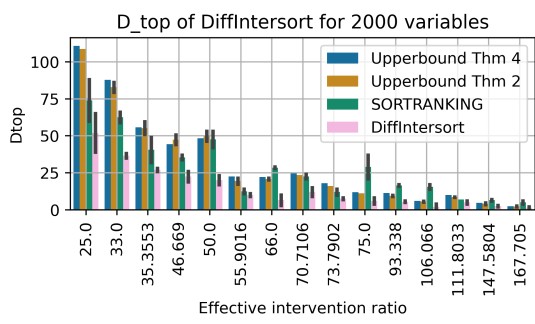 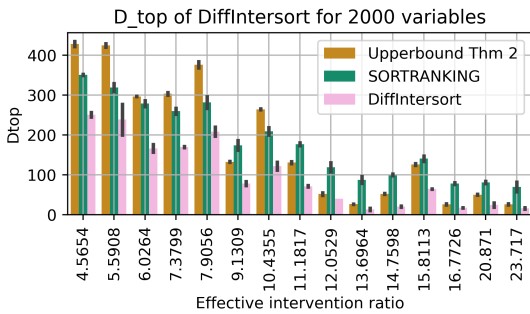

(a) Simulation ER with 2000 variables   (b) Simulation SF with 2000 variables

Figure 1: Simulation and comparison between the bounds of Thm 2 and 4 of Chevalley et al. (2024) for Erdős-Rényi (ER, left) and scale-free networks (SF, right) for 2000 variables. We compare the causal order obtained by maximizing our proposed DiffIntersort score and the output of SORTRANKING. For each setting, we draw 1 graphs per setting, following a ER distribution with a probability of edges per variable $p_e$ in $\{0.0001, 0.00005, 0.00002\}$ and following a Barabasi-Albert SF distribution, with average edge per variable in $\{1, 2, 3\}$. A setting is the tuple $(p_{int}, p_e)$, where $p_e = \frac{2\mathrm{E}(\#edges)}{d(d-1)}$ for the SF distribution. Then, for each graph, we run the algorithm on 1 configuration, where each configuration corresponds to a draw of the targeted variables following $p_{int}$. We have $p_{int} \in \{0.25, 0.33, 0.5, 0.66, 0.75\}$. The settings are ordered on the x-axis following what is called the effective intervention ratio $\frac{p_{int}}{\sqrt{p_e}}$ (Chevalley et al., 2024).

generated using the SERGIO tool (Dibaeinia & Sinha, 2020), utilizing the code provided by Lorch et al. (2022) (MIT License, v1.0.5). We also test our method on neural network functional data following the setup of Brouillard et al. (2020), using the implementation from Nazaret et al. (2023) (MIT License, v0.1.0). To assess the impact of interventions, we vary the ratio of intervened variables in the set $25\%, 50\%, 75\%, 100\%$. All datasets are standardized based on the mean and variance of the observational data to eliminate the Varsortability artifact identified by Reisach et al. (2021). For the linear and RFF domains, the noise distribution is chosen uniformly at random from the following options: uniform Gaussian (noise scale independent of the parents), heteroscedastic Gaussian (noise scale functionally dependent on the parents), and Laplace distribution. In the neural network domain, the noise distribution is Gaussian with a fixed variance. We conduct experiments on 10 simulated datasets for each domain and each ratio of intervened variables. The observational datasets contain 5,000 samples, and each intervention dataset comprises 100 samples, mirroring the sample sizes typically found in real single-cell transcriptomics studies (Replogle et al., 2022).

We compare the performance of DiffIntersort and SORTRANKING (Chevalley et al., 2024) as measured by the top order divergence $D_{top}$ on 100 variables in fig. 2. For the DiffIntersort score and the Intersort score, we use the same parameters as in Chevalley et al. (2024): $\epsilon = 0.3$ for linear, RFF and NN data, and $\epsilon = 0.5$ for GRN data, and $c = 0.5$. We use the Wasserstein distance (Villani et al., 2009) for the statistical metric. Results for 10 and 30 variables, additionally compared to Intersort, can be found in the appendix in fig. 7. As can be observed, the performance of the two algorithms is close. This demonstrates that the optimizing DiffIntersort can be solved at scale using continuously differentiable optimization also on realistic synthetic data.

We now evaluate our causal discovery method on synthetic datasets generated using linear structural equation models (SEMs), gene regulatory network (GRN) models, random Fourier features (RFF) models, and neural network (NN) models as presented previously. For each model type, we consider variable sizes of 10, 30, and 100 to assess scalability and performance across different problem dimensions. We use two evaluation

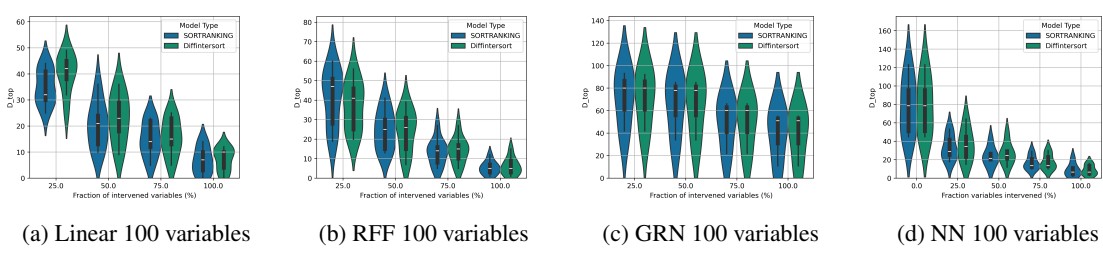

(a) Linear 100 variables     (b) RFF 100 variables     (c) GRN 100 variables     (d) NN 100 variables

Figure 2: Top order diverge scores (lower is better) assessing the quality of the derived causal order, comparing our method based on the DiffIntersort score to SORTRANKING on 100 variables, for various types of data.

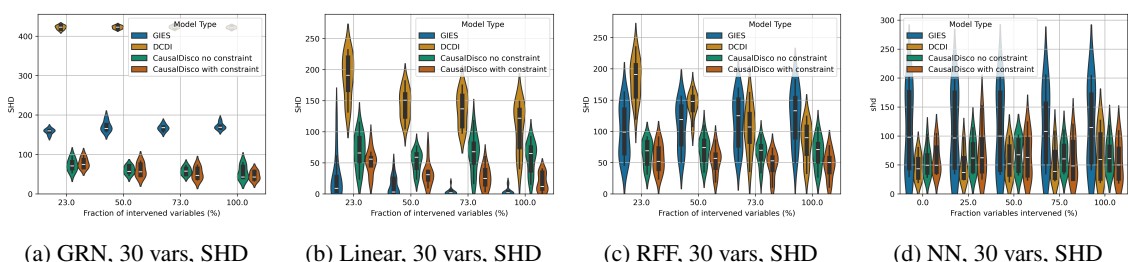

(a) GRN, 30 vars, SHD     (b) Linear, 30 vars, SHD     (c) RFF, 30 vars, SHD     (d) NN, 30 vars, SHD

Figure 3: Comparison of SHD (lower is better) for GRN, Linear, RFF, and Neural Network data with varying numbers for 30 variables. Our method (CausalDisco with and without constraint) achieves lower SHD values compared to baseline methods on GRN and RFF data. GIES outperforms on the linear data and DCDI performs slightly better.

metrics: Structural Hamming Distance (SHD) (Tsamardinos et al., 2006) and Structural Intervention Distance (SID) (Peters & Bühlmann, 2015) to compare the inferred graphs with the true causal graphs. We compared to two baselines, namely GIES (Hauser & Bühlmann, 2012) and DCDI (Brouillard et al., 2020). We note that those two baselines do not scale to 100 variables. For our model, we compare the performance of our proposed causal discovery model with and without the DiffIntersort constraint. We present the results for the SHD metrics at 30 variables in fig. 3. The results for 10 and 30 variables for SHD can be found in the appendix in fig. 8. The results for SID can be found in fig. 9l in the appendix.

As can be seen, the DiffIntersort constraint is consistently beneficial in terms of performance on both metrics, for all types of data and at all considered scales. This comparison validates the usefulness of inducing the interventional faithfulness inductive bias to a causal models via the DiffIntersort score. We expect that this approach may be applicable to other causal tasks of interest, in settings where a large set of single variable interventions are available. Compared to baselines, our model outperforms on the GRN and RFF data. GIES is the best model on linear data, and DCDI has a slightly better performance on NN data. GIES and DCDI do not scale to 100 variables but we would expect the results to be the same, as our algorithm has an F1 score that is almost unaffected by the number of variables (see fig. 4). The results on the F1 score also shows the robustness of our causal discovery model with the DiffIntersort constraint to the number of variables.

## 5 CONCLUSION

In this work, we addressed the scalability and differentiability limitations of Intersort, a score-based method for discovering causal orderings using interventional data. By reformulating the Intersort score through

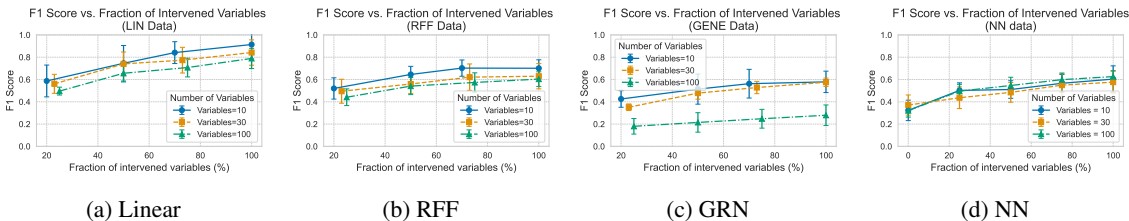

|         |         |         |         |
|---------|---------|---------|---------|
| (a) Linear | (b) RFF | (c) GRN | (d) NN |

Figure 4: F1 score of our algorithm with DiffIntersort constraint for the four considered data types over the fraction of intervened variables for 10, 30 and 100 variables. As can be observed, apart for the GRN data, the performance is consistent across the scale of the number of variables as there is no major drop in performance at 100 variables compared to 10 and 30 variables.

differentiable sorting and ranking techniques—specifically utilizing the Sinkhorn operator—we enabled the scalable and differentiable optimization of causal orderings. This reformulation allows the Intersort score to be integrated as a continuous regularizer in gradient-based learning frameworks, facilitating its use in downstream causal discovery tasks. Our proposed approach not only preserves the theoretical advantages of Intersort but also significantly improves its practical applicability to large-scale problems. Empirical evaluations demonstrate that incorporating the differentiable Intersort score into a causal discovery algorithm leads to superior performance compared to existing methods, particularly in complex settings involving non-linear relationships and large numbers of variables. The algorithm exhibits robustness across various data distributions and noise types, effectively scaling with increasing data size without compromising performance. By bridging the gap between interventional faithfulness and differentiable optimization, our work opens new avenues for integrating interventional data into modern causal machine learning pipelines. This advancement holds promise for a wide range of applications dealing with large covariate sets where understanding causal relationships is crucial, such as genomics, neuroscience, environmental, and social sciences.

While our approach enhances the scalability and differentiability of causal discovery using interventional data, several avenues for future research remain. One potential direction is to integrate our differentiable Intersort score into more complex models, such as deep neural networks, to further improve causal discovery in high-dimensional and highly non-linear settings. Another promising area is the application of the differentiable Intersort approach to real-world datasets in domains like genomics or healthcare, where large-scale interventional data are increasingly available.

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

## 6 APPENDIX

### 6.1 PROOFS

*Proof of Theorem 1.* First, let us recall that we have $\boldsymbol{p} \in \mathbb{R}^d$, and $\pi \in \{0, 1, \ldots, d\}^d$, where $\forall i, j \in \{0, 1, \ldots, d\}, \pi_i \neq \pi_j$. We thus trivially have that any permutation $\pi$ can be represented by a potential $\boldsymbol{p}$, by $\boldsymbol{p}_i = -\pi_i \forall i \in \{0, 1, \ldots, d\}$. We now have to prove that if $\pi \in \Pi$, then the corresponding potential $\boldsymbol{p}_\pi \in \mathbb{P}$. Let $s = \max_\pi S(\pi, \epsilon, D, \mathcal{I}, P_X^{\mathcal{C},(\emptyset)}, \mathcal{P}_{int}, c)$ be the maximum achievable score. The sum of the score is over the elements of $\mathbf{D}_{ij}$ where $\pi_i < \pi_j$. For all these pairs of indices, we also have that $p_{\pi_i} > p_{\pi_j}$, and thus for all those pairs, we also have $(\text{Step}(\text{grad}(\boldsymbol{p}_\pi)))_{ij} = 1$. This exactly corresponds to the elements that are non-zero and thus contribute to the sum in eq. (3). Thus we have that $S(\boldsymbol{p}_\pi) = s$, and as such $\boldsymbol{p}_\pi \in \mathbb{P}$, which concludes the proof. □

### 6.2 ADDITIONAL EXPERIMENTS

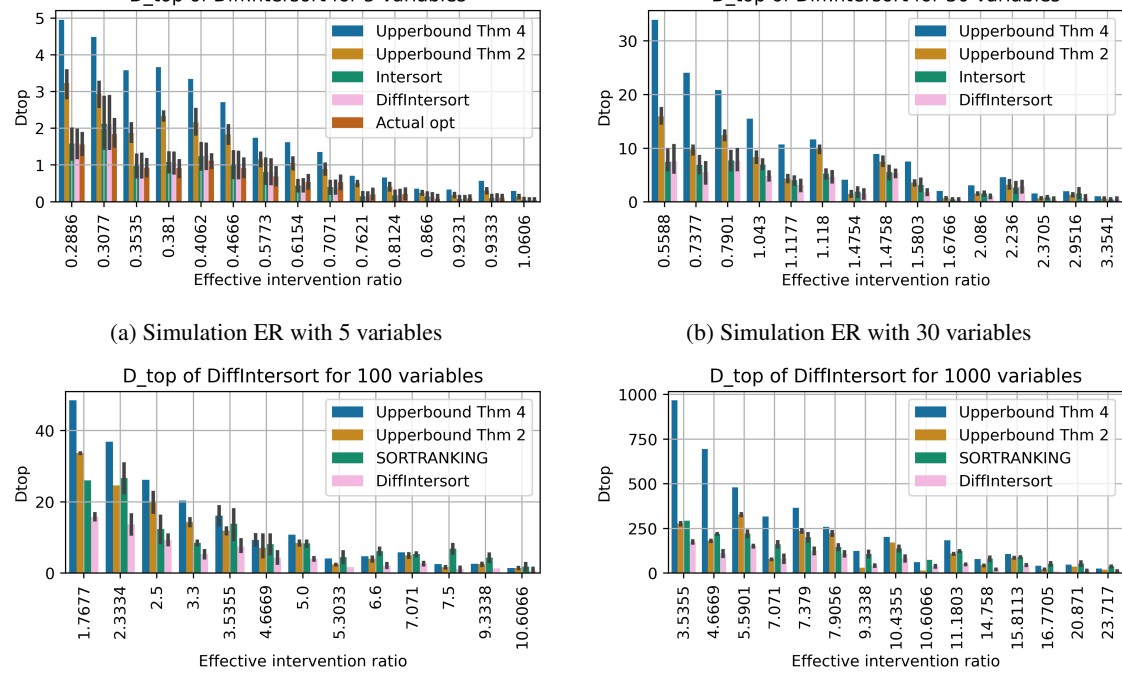

Figure 5: Comparison of performance on simulated ER graphs in terms of $D_{top}$ divergence between the two bounds of (Chevalley et al., 2024), DiffIntersort, Intersort and SORTRANKING. For each setting, we draw multiple graphs, where a setting is the tuple $(p_{int}, p_e)$. Then, for each graph, we run the algorithm on multiple configurations, where a configuration corresponds to a set of intervened variables following $p_{int}$. We have $p_{int} \in \{0.25, 0.33, 0.5, 0.66, 0.75\}$ for all scales. For 5 variables, we have $p_e \in \{0.5, 0.66, 0.75\}$. For 30, we have $p_e \in \{0.05, 0.1, 0.2\}$. For 1000 variables, we have $p_e \in \{0.005, 0.002, 0.001\}$. For 20000 variables settings, we have $p_e \in \{0.0001, 0.00005, 0.00002\}$. Those edge probabilities approximately correspond to an average of 1, 2 or 3 edges per variable.

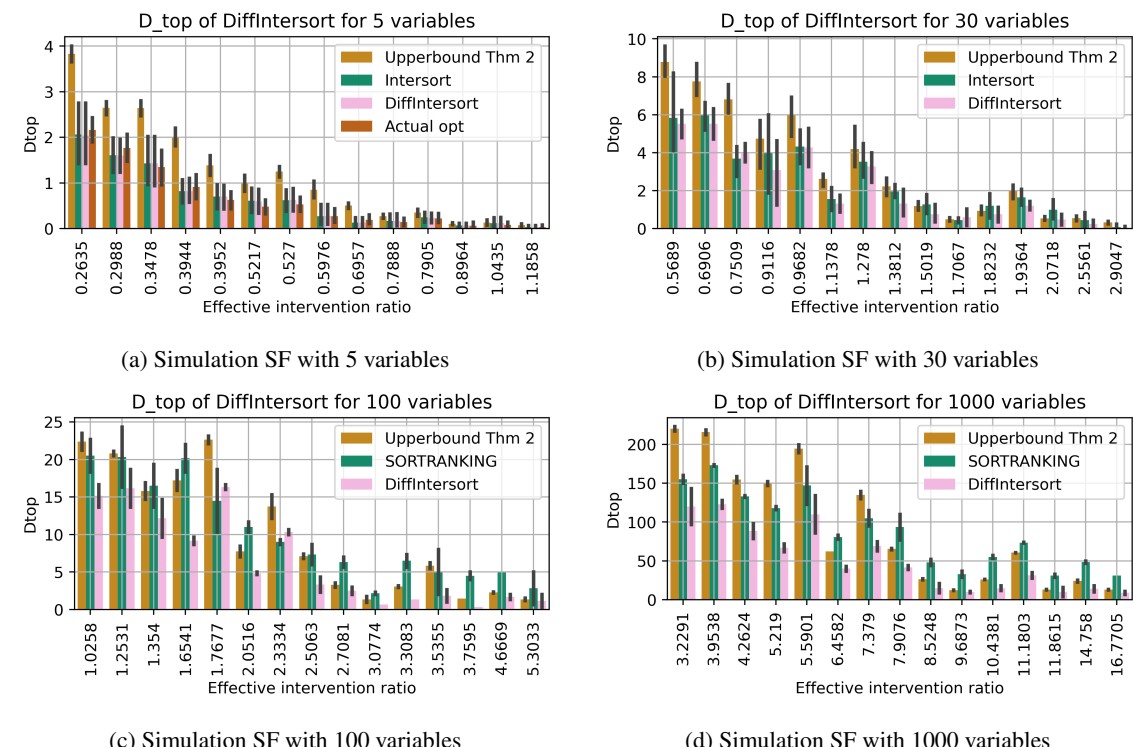

(a) Simulation SF with 5 variables

(b) Simulation SF with 30 variables

(c) Simulation SF with 100 variables

(d) Simulation SF with 1000 variables

Figure 6: Comparison of performance on simulated SF graphs in terms of $D_{top}$ divergence between the two bounds of (Chevalley et al., 2024), DiffIntersort, Intersort and SORTRANKING. For each setting, we draw multiple graphs, where a setting is the tuple $(p_{int}, p_e)$. The networks follow a Barabasi-Albert SF distribution, with average edge per variable in $\{1, 2, 3\}$. A setting is the tuple $(p_{int}, p_e)$, where $p_e = \frac{2\mathrm{E}(\#edges)}{d(d-1)}$. Then, for each graph, we run the algorithm on multiple configurations, where a configuration corresponds to a set of intervened variables following $p_{int}$. We have $p_{int} \in \{0.25, 0.33, 0.5, 0.66, 0.75\}$ for all scales.

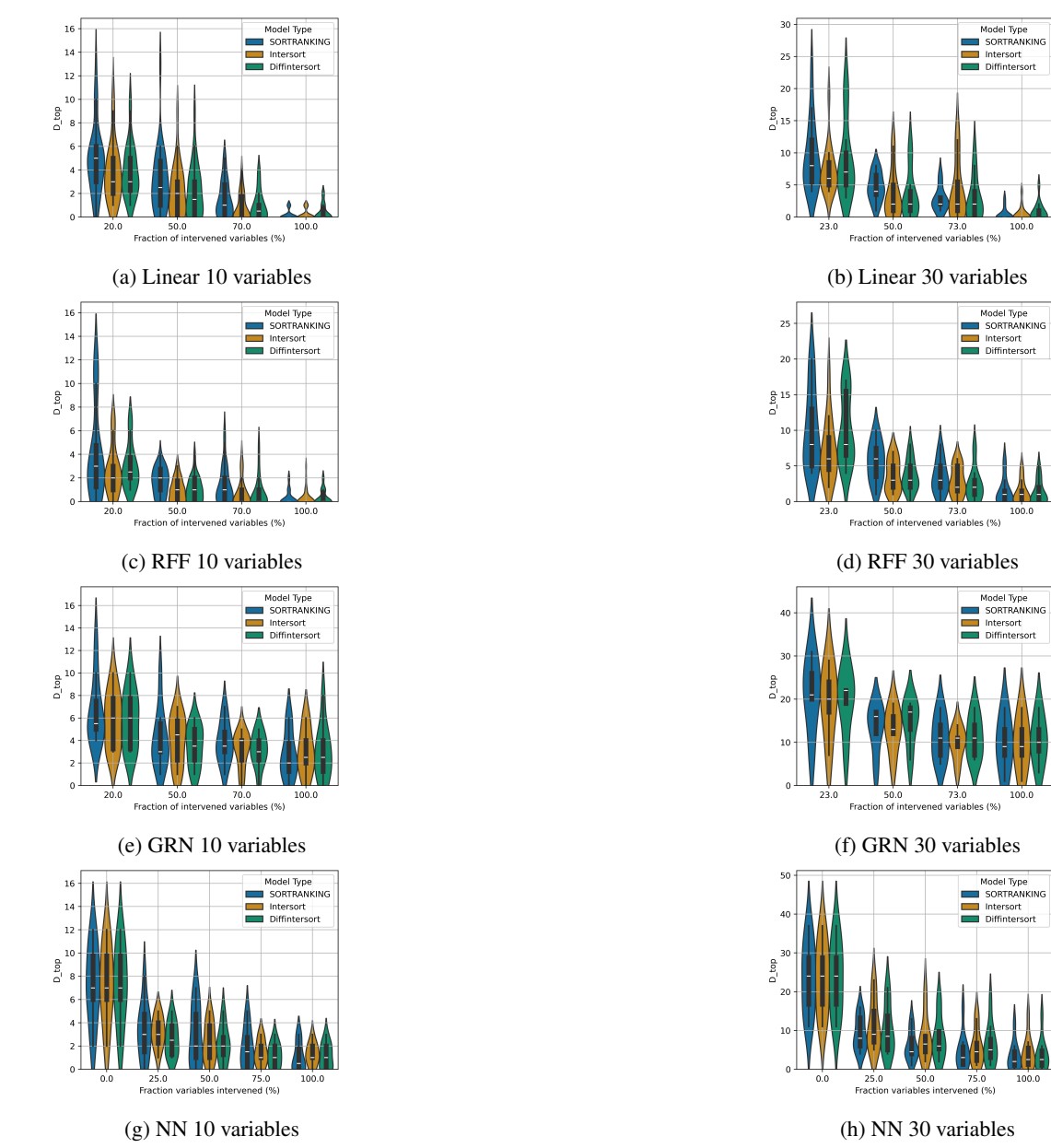

Figure 7: Top order diverge scores (lower is better) assessing the quality of the derived causal order, comparing our method based on the DiffIntersort score to SORTRANKING and Intersort on 10 and 30 variables, for various types of data.

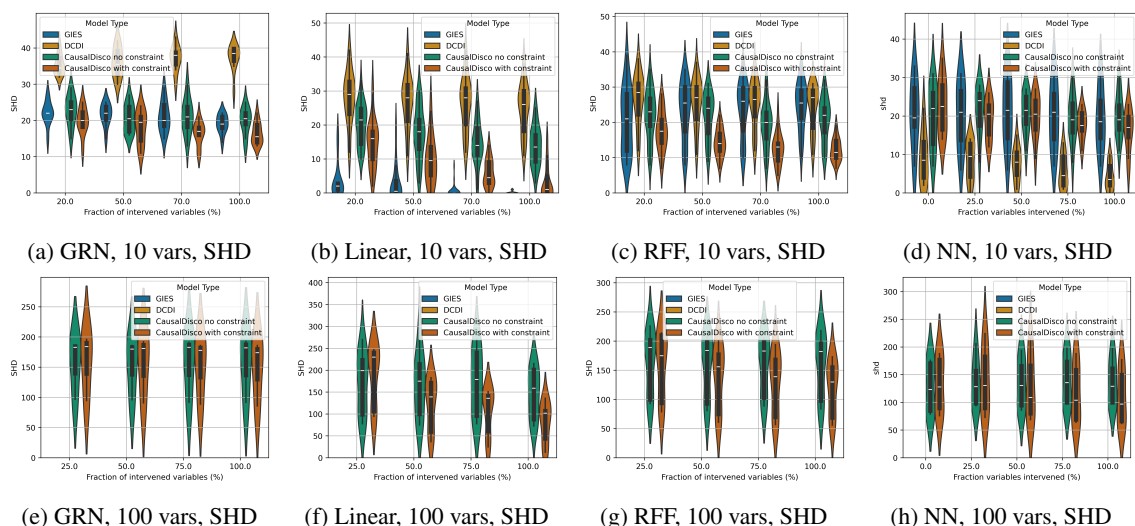

(a) GRN, 10 vars, SHD    (b) Linear, 10 vars, SHD    (c) RFF, 10 vars, SHD    (d) NN, 10 vars, SHD

(e) GRN, 100 vars, SHD    (f) Linear, 100 vars, SHD    (g) RFF, 100 vars, SHD    (h) NN, 100 vars, SHD

Figure 8: Comparison of Structural Hamming Distance (SHD) and Structural Intervention Distance (SID) for Gene, Linear, RFF, and Neural Network models with varying numbers of variables. Our method (DiffIntersort) consistently achieves lower SHD and SID values compared to baseline methods, indicating more accurate causal graph recovery.

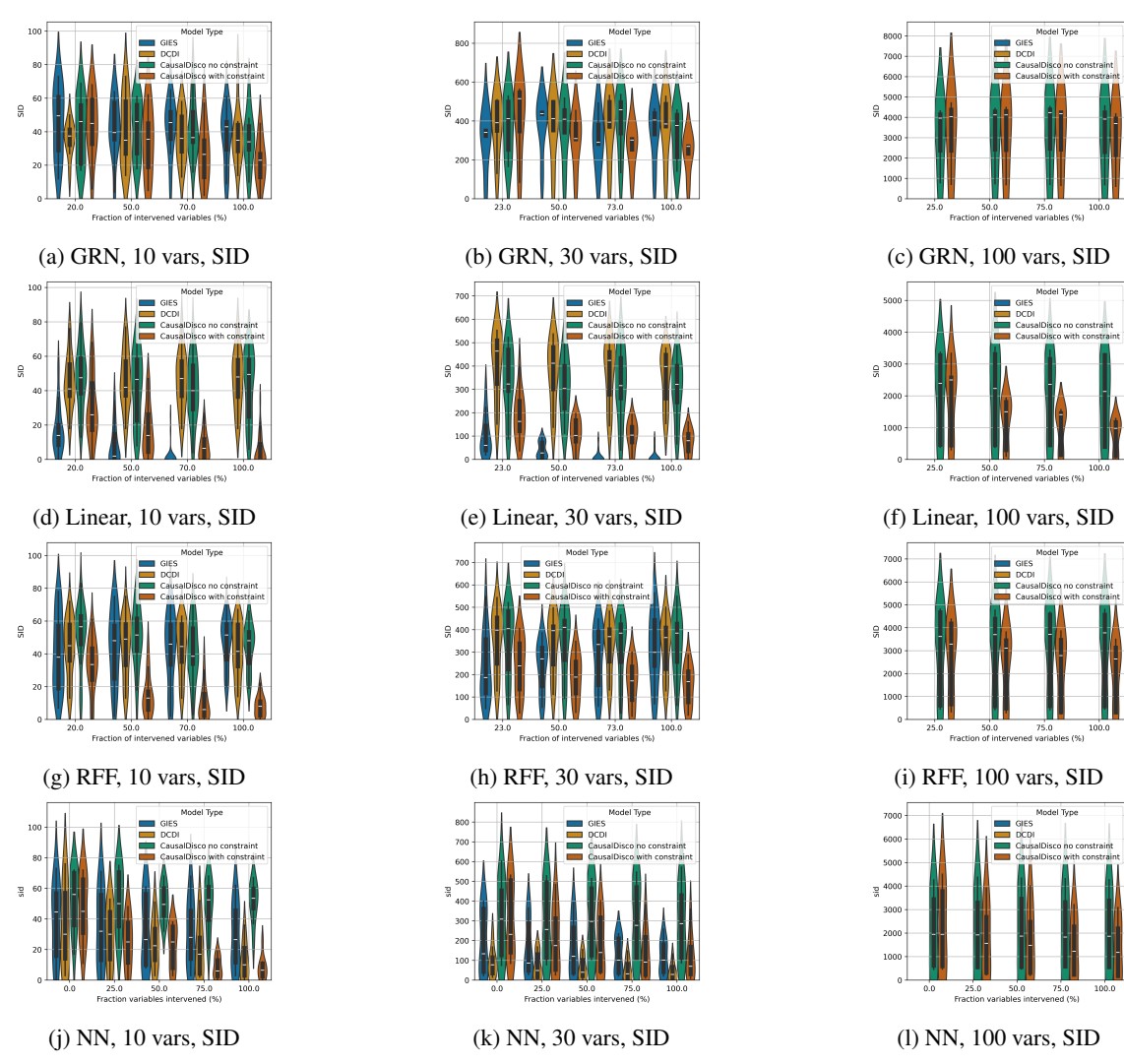

(a) GRN, 10 vars, SID     (b) GRN, 30 vars, SID     (c) GRN, 100 vars, SID

(d) Linear, 10 vars, SID     (e) Linear, 30 vars, SID     (f) Linear, 100 vars, SID

(g) RFF, 10 vars, SID     (h) RFF, 30 vars, SID     (i) RFF, 100 vars, SID

(j) NN, 10 vars, SID     (k) NN, 30 vars, SID     (l) NN, 100 vars, SID

Figure 9: Comparison SID (lower is better) for GRN, Linear, RFF, and Neural Network models with varying numbers of variables. Our method (DiffIntersort) consistently achieves lower SHD and SID values compared to baseline methods, indicating more accurate causal graph recovery.

