# OpenReview forum: "Efficient Differentiable Discovery of Causal Order"
_ICLR.cc/2025/Conference — Submitted to ICLR 2025_

### Official Review · Reviewer_Qrgp · 2024-10-29

**Soundness:** 2
**Presentation:** 1
**Contribution:** 3
**Rating:** 5
**Confidence:** 4

**Summary:**

In this paper, the authors propose a modification of the Intersort algorithm from Chevalley et al., 2024. Intersort adopts a scoring function, that, under interventional faithfulness assumption and access to single variable interventions for each node, if maximized yields the correct causal order. The scoring function measures the fit of the casual order based on its compatibility with interventional data. The downside of this method is that it does not scale to graphs with many nodes (up to 100, according to the authors): this work introduces Diffintersort as an alternative that is differentiable and proposes to scale to thousands of nodes.

**Strengths:**

The problem of differentiable causal discovery is of great interest given that, as mentioned by the authors, causal discovery algorithms that can be trained by backpropagation can be used as components of deep learning pipelines. The main contribution of this work is the definition of an alternative score (alternative to that of Intersort) that is differentiable and ensures recovery of an optimal causal order when the score is maximized. Empirically, Diffintersort appears to be scaling to large graphs, with impressive performance when the number of interventions increases. I believe this algorithm can make a good contribution to the causal discovery community

**Weaknesses:**

## Main points

1. One main issue I found is that even though *efficiency* is in the title, its empirical and theoretical analysis is not treated in the paper. The novel contribution of this work consists of the algorithmic part, which is a variation of Intersort (as the theory heavily builds on Chevalley, 2024, and Annadani, 2023): however, for this to be considered a sufficiently relevant contribution, simply stating that *Intersort does not scale beyond 100 variables* (L314), whereas DiffIntersort can scale to 2000 variables, is not enough. This must be theoretically and empirically analyzed. In detail:
    1. An analysis of the algorithmic complexity of DiffIntersort is missing, and this should be done in comparison to an analysis of the algorithmic complexity of Intersort
    2. There is no trace of the execution times of DiffIntersort in the experiments, under different conditions (especially varying the number of nodes, as I believe this is what the authors mean by *efficiency*), and in comparison with the execution times of Intersort.
2. The experimental section has some obscure points and limitations in its reproducibility
    1. It is not mentioned what optimization algorithm is used for the experiments.
    2. It is unclear how hyperparameters are set. The authors simply say that they choose $\epsilon = 0.3$, $\gamma = 0.5$, there is no mention of how $\lambda_1, \lambda_2$ parameters of the loss are set, and the same goes for $c$.
    3. Some of the proposed mechanisms are unclear or need more discussion. The authors say that they use neural networks to generate nonlinear mechanisms as in Brouillard et al., 2020: in that paper, the authors define different classes of structural causal models (with and without additive noise) using neural networks. Which one are they referring to? Moreover, the discussion of random Fourier features generation is overlooked: I believe it would be beneficial to provide some reference to the topic and ideally some discussion in the appendix. Finally, there is no mention of how weights are sampled for the linear mechanisms. In general, please discuss in much greater detail the data generation.

## Other points ##
1. In the nonlinear case (potentially with non-additive noise) it is not discussed how $\hat X_i$ in the loss formulation is computed. The authors detail computations for the linear case as an example, using masking on the weight matrices that ensures acyclicity, yet it is unclear how this is achieved in the nonlinear case.
2. The authors claim that they prove DiffIntersort inference abilities on a variety of noise terms, yet they only adopt Gaussian and Laplacian distribution. This is quite a limited choice: I suggest adding more distributions, or you could also consider densities as nonlinear transformations parametrized by neural networks as in Montagna et al., 2023 (https://arxiv.org/abs/2310.13387)

**Questions:**

## Main ##
As recently discussed by Ng et al., 2024 (https://proceedings.mlr.press/v236/ng24a/ng24a.pdf), non-convexity of the loss might constitute the biggest challenge to differentiable causal discovery methods. What are the authors' thoughts on this? This should be a subject of discussion in the paper.

## Others ##
1. Do the authors assume causal sufficiency? Do they assume independence of the noise terms?
2. The experiments of Figure 1 are carried out on very sparse graphs, in the ER setting: specifically, authors sample graphs with 2000 nodes, and the expected number of edges {200, 100, ~40} corresponding to $p_e$ = {1e-4, 5e-5, 2e-5}. I would ask to consider experiments in much denser settings.

---

### Official Review · Reviewer_dWWR · 2024-11-02

**Soundness:** 2
**Presentation:** 3
**Contribution:** 1
**Rating:** 3
**Confidence:** 3

**Summary:**

The authors introduce DiffIntersort as a differentiable approximation of the existing Intersort algorithm (Chevalley et al., 2024). By minimizing a score-function, Intersort first fits from data a complete order between variables that is then refined by exploring the space of permutations. In this paper, the authors define the score as a differentiable function using the Sinkhorn operator and then propose to jointly minimize this score and a loss function on the observations. DiffIntersort can handle interventional samples, and it is thus compared with GIES and DCDI on synthetic data and simulated but plausible gene regulatory networks.

**Strengths:**

The development of scalable approaches for causal discovery is valuable and has meaningful applications in important contexts, such as genomics. The proposed algorithm performs better than the other baselines on some scenarios (gene-regulatory and random Fourier features) and on par with them on the remaining (linear SEMs and neural network simulated data).

**Weaknesses:**

The paper does not reference in the related works significant works of recent literature. First, the decomposition of the matrix to ensure acyclicity without constraints (Eq. 3) strongly resembles a parameterization introduced by NO-CURL using a ReLU function (Yu et al. 2021) and by COSMO using a smooth acyclic orientation (Massidda et al. 2024), both in the observational context. Overall, the algorithm has even more similarities with BayesDAG (Annadani et al. 2023) and DP-DAG (Charpentier et al. 2022). In particular, DP-DAG also supports interventional data and uses the Gumbel-Sinkhorn operator and the Hungarian algorithm to get a differentiable representation of a permutation. Overall, the paper should have a more comprehensive related work section and compare experimentally with other differentiable approaches for interventional data.

The authors motivate their differentiable extension of Intersort to avoid its substantial computational cost. It would aid the presentation if the authors could report it and, more importantly, it could improve the contribution if the authors actually reported what the computational complexity of DiffIntersort is. On this matter, it's worth recalling that DP-DAG has quadratic complexity on the number of nodes. Furthermore, I would expect in a paper focused on the *efficiency* of the algorithm to have at least one comparison of the execution time of the algorithm and its competitors.

Also, please note that your solution is incorrectly, I guess, referred to as "causaldisco" in Figure 3.

*References* in alphabetical order.
- Annadani, Yashas, et al. "Bayesdag: Gradient-based posterior inference for causal discovery." Advances in Neural Information Processing Systems 36 (2023): 1738-1763.
- Charpentier, Bertrand, Simon Kibler, and Stephan Günnemann. "Differentiable DAG Sampling." International Conference on Learning Representations.
- Massidda, Riccardo, et al. "Constraint-Free Structure Learning with Smooth Acyclic Orientations." The Twelfth International Conference on Learning Representations.
- Yu, Yue, et al. "DAGs with no curl: An efficient DAG structure learning approach." International Conference on Machine Learning. Pmlr, 2021.

**Questions:**

- How does the computational complexity of DiffIntersort compare to existing differentiable approaches for interventional data such as DP-DAG?
- How do you handle $p_i=p_j$? Does that connect the nodes bidirectionally or does it remove the edge altogether?
- What is your wall time when you claim that the baselines do not scale over 100 variables?

---

### Official Review · Reviewer_DcnJ · 2024-11-03

**Soundness:** 2
**Presentation:** 2
**Contribution:** 1
**Rating:** 3
**Confidence:** 4

**Summary:**

The paper considers the problem of identifying the causal ordering from interventional data. The proposed method is based on an existing score-based algorithm named "Intersort". Different from Intersort that requires score evaluation over permutations, this work uses differentiable sorting and ranking techniques, namely the Sinkhorn operator, to enable gradient-based optimization for causal discovery. The proposed approach is claimed to be efficient and scalable.

**Strengths:**

1. The experimental results is relatively comprehensive.

2. The introduction of differentiable operator is novel to me and may be beneficial to other permutation based methods.

**Weaknesses:**

1. The motivation of the work is unclear:
   - What is being discovered? In section 3.3 the authors say "Our goal is to recover the causal structure and ordering of the variables from both observational and interventional data", but in the title and elsewhere just the causal ordering is emphasized. So which is exactly the objective?
   - Note that with the interventional data available the whole graph's equivalence class can be identified and the complexity for such whole structure learning is almost the same -- given a causal ordering, the exact DAG from its CPDAG (or smaller equivalence class due to interventions) can be identified trivially, using the local Markov property. Therefore, I doubt that the causal ordering learning in this setting is  not a necessary problem.

2. The identifiable guarantee of the work needs more justification:
   - What are exactly all the assumptions needed in this work? I only see Interventional Faithfulness. Does it mean that this work, together with the score used and the distance function used, is fully nonparametric? No assumptions on e.g., the functional model, the noise distributions (which other differential methods usually require) are required? And what are the hidden assumptions under the distance measure, e.g., is it meaningful to compare the distances?
   - For the identifiability of the causal order itself, what is needed? The authors says they "derive the causal order of the variables when many single-variable interventions are available", but how `many' is many? That a causal ordering can be fixed yields that the whole DAG structure can be identified (from its CPDAG or smaller equivalence class). Then do the authors implicitly assume this?
   - Overall, since Intersort may not appear as a well-known method for now, some background over that algorithm should be given, before providing any details of implementation or efficiency improvements in the paper.

3. The presentation can be improved. E.g., the definition 2 (top order divergence) serves only as a metric of evaluation, and should not be put at the beginning of the method part to distract readers.

4. More detailed literature review on causal discovery from interventional data, and on the differentiable ordering learning is expected.

**Questions:**

See above.

---

### Official Review · Reviewer_8dBB · 2024-11-04

**Soundness:** 2
**Presentation:** 2
**Contribution:** 3
**Rating:** 5
**Confidence:** 3

**Summary:**

This paper focuses on a differentiable version of the Intersort score, a score for identifying the topological ordering of a causal graph via interventional data (Chevalley et al. 2024). This proceeds by rewriting the Intersort score to be parameterized by a potential, and applying tools used for gradient-based optimization of permutations, resulting in DiffIntersort. A differentiable score for topological ordering allows for a novel regularization scheme in score-based causal discovery, and a novel method CausalDisco is presented, which optionally includes DiffIntersort as a regularization term, and is tested empirically against DCDI and GIES. It is shown empirically that DiffIntersort provides scores generally consistent with Intersort with significantly less computational cost. CausalDisco provides mixed results, outperforming benchmark methods in two of four tested settings, but regularization with DiffIntersort is shown to consistently benefit CausalDisco.

**Strengths:**

- I find the idea of regularizing based on topological ordering very interesting, especially because this can cause many downstream errors. Using a differentiable version of Intersort as a regularizer seems a fitting and effective way to realize this idea. It is also clever to derive a score function which uses learned causal orders to avoid acyclicity constraints.
- Empirically and theoretically, the results of DiffIntersort seem close to Intersort, which is reassuring for the differentiable relaxation.

**Weaknesses:**

**On the Presentation of the Main Ideas**

I find the novelty in Intersort -> DiffIntersort somewhat limited; I recognize the use of a potential is nontrivial, but this is still mostly just combining Chevalley et al. (2024) and Annadani et al. (2023) in a straightforward way. I find this particularly unfortunate because I think regularization based on ordering to be a much more novel and interesting idea. While subjective, I think the paper would benefit from focusing on order-based regularization, with DiffIntersort as a way to achieve that (rather than focusing on DiffIntersort, with order-based regularization an application thereof).

Note that the effects of this propagate to the experimental section: for example, I do not see any experiments considering the effect of varying $\lambda_2$, which is an interesting ablation from the perspective of regularization. (Indeed, I cannot seem to find the actual value of $\lambda_2$ used -- please let me know if I am missing it). Causal discovery with interventional data has also been an increasingly popular topic, and I imagine DiffIntersort could be used to regularize a more popular objective function.

**On Reproducibility**

Code is provided to compute DiffIntersort scores, which is great. However, it doesn't seem like there is code available for CausalDisco, or for the experiments included in the paper. Especially since the details in the manuscript regarding CausalDisco are somewhat sparse (e.g.., hyperparmeters used for regaularization), code would be very helpful in ensuring the reproducibility of results.

**On the "DiffIntersort Constraint"**

I have doubts about the presentation of "CausalDisco with constraint." A constraint only appears in Eq. (8), and it is likely more appropriate to present as a bilevel optimization problem. Moreover, the actual CausalDisco loss is presented as an unconstrained regularization problem, solutions of which are (a priori) not generally equivalent to the corresponding bilevel problem.

**Minor Note on Naming**

The closest ideas to the proposed CausalDisco are likely those of Reisach et al. (2021) and Reisach et al. (2023). The library developed for their work is called "CausalDisco", which can be a source of possible confusion.

## Minor Typos
- Line (220) Kuhn (1955) should use `\citep{}`
- (Line 316) Albert & Barabasi (2002) should use `\citep{}`
- (Line 408) I think the reference should be to simply Figure 9.

**Questions:**

- As mentioned in weaknesses, constrained optimization only appears in Eq. (8), and the experiments and loss function proposed do not solve the constrained problem. Is there a particular reason to consider this constrained problem, instead of focusing instead on regularization from the start?
- What is the role of regularization strength? How is $\lambda_2$ determined, and what is the effect of varying it?
- I also am not sure I quite understand the final model used; a linear model is given in Equation (10), but loss functions are specified in terms of $\ell(\mathbf{x}_i, \hat{\mathbf{x}}_i; \theta)$; is the linear model used in all experiments, with $\theta = \\\{ \mathbf{W}, \mathbf{b} \\\} $?

## References

Reisach, A., Seiler, C., & Weichwald, S. (2021). Beware of the simulated DAG! Causal discovery benchmarks may be easy to game. Advances in Neural Information Processing Systems, 34, 27772-27784.

Reisach, A., Tami, M., Seiler, C., Chambaz, A., & Weichwald, S. (2024). A scale-invariant sorting criterion to find a causal order in additive noise models. Advances in Neural Information Processing Systems, 36.

Chevalley, M., Schwab, P., & Mehrjou, A. (2024). Deriving Causal Order from Single-Variable Interventions: Guarantees & Algorithm. arXiv preprint arXiv:2405.18314.

**Details Of Ethics Concerns:**

Annadani et al. (2023) is cited, but there is still a potentially unacceptable amount of paraphrasing when explaining the Sinkhorn operator. In particular, lines 205-217 on page 5 of this submission are essentially sentence-by-sentence identical to the top of page 5 of Annadani et al. (2023), occasionally replacing a word with a synonym. This does not seem intended and I am not aware of other instances in the manuscript, but still may be considered improper paraphrasing.

## References
Annadani, Y., Pawlowski, N., Jennings, J., Bauer, S., Zhang, C., & Gong, W. (2023). BayesDAG: Gradient-based posterior sampling for causal discovery. arXiv preprint arXiv:2307.13917.

---

### Meta-Review · Area_Chair_ykft · 2024-12-20

**Metareview:**

This paper introduces a differentiable version of the Intersort algorithm, and proposes CausalDisco, a causal discovery method using DiffIntersort for regularization. Reviewers found the scalability and differentiability promising but raised concerns about limited theoretical and empirical analysis, unclear objectives, incomplete implementation details, and missing comparisons with relevant methods. An ethics concern was also flagged regarding potential improper paraphrasing. While promising, the paper requires significant revisions to clarify its contributions and address these issues.

**Additional Comments On Reviewer Discussion:**

The authors did not provide any rebuttals.

---

### Decision · Program_Chairs · 2025-01-22

Reject